# Electromyographic biofeedback therapy for improving limb function after stroke: A systematic review and meta-analysis

Rui Wang[1,2☯], Shuangshuang Zhang[2☯], Jie Zhang[2], Qifeng Tong[2,3], Xiangming Ye[2], Kai Wang[4]*, Juebao Li[2]*

1 Department of The Second Affiliated Hospital of Anhui University of Chinese Medicine, Hefei, China, 2 Center for Rehabilitation Medicine, Rehabilitation & Sports Medicine Research Institute of Zhejiang Province, Department of Rehabilitation Medicine, Zhejiang Provincial People's Hospital, Affiliated People's Hospital, Hangzhou Medical College, Hangzhou, Zhejiang, China, 3 College of Rehabilitation, Zhejiang Chinese Medical University, Hangzhou, China, 4 Department of Neurology, The Second Affiliated Hospital of Xuzhou Medical University, Xuzhou, Jiangsu, China

☯ These authors contributed equally to this work.
* lijuebao@126.com (JL); wk1982405@163.com (KW)

**Data Availability Statement:** All relevant data are within the paper and its Supporting information files.

## Abstract

### Background

Upper and lower limb impairment is common after stroke. Electromyographic biofeedback therapy is a non-invasive treatment, and its effectiveness in functional rehabilitation of the limb after stroke still remains uncertain.

### Objective

The objective of this study was to evaluate whether electromyographic biofeedback can improve upper and lower limb dysfunction in stroke patients.

### Methods

PubMed, Embase, Cochrane Library, and Physiotherapy Evidence Database (PEDro) were searched from inception to 1st May 2022. Inclusion criteria were randomized controlled clinical trials of electromyographic biofeedback therapy interventions reporting changes in upper and lower limb function in post-stroke patients. Data were extracted by two independent reviewers and pooled in random-effects models using Review manager (RevMan) software.

### Results

Our analyses included 10 studies enrolling a total of 303 participants. Electromyographic biofeedback therapy can effectively improve limb function after stroke (standardized mean difference [SMD], 0.44; 95% confidence interval [CI], 0.12–0.77; P = 0.008) and in subgroup analyses, the effect sizes of short-term effect (SMD, 0.33; 95% CI, 0.02–0.64; P = 0.04) was significant, but the long-term was not (SMD, 0.61; 95% CI, -0.11–1.33; P = 0.10). In addition, Electromyographic biofeedback therapy can improve the active range of motion of shoulder

**Funding:** Juebao Li received award; The Natural Science Foundation of Zhejiang Province (Grant No. LGF20H170012). Include this sentence at the end of your statement: The funders had no role in study design, data collection and analysis, decision to publish, or preparation of the manuscript.

**Competing interests:** The authors have declared that no competing interests exist.

(SMD, 1.49; 95% CI, 2.22; P<0.0001) and wrist joints (SMD, 0.77; 95% CI, 0.13–1.42; P = 0.02) after stroke.

## Conclusion

In this meta-analysis, electromyographic biofeedback therapy intervention can improve upper and lower limb function in patients with stroke. Short-term (less than one month) improvement after electromyographic biofeedback therapy was supported, while evidence for long-term (more than one month) benefits was lacking. Range of motion in the gleno-humeral and wrist joints were improved. Stronger evidence for individualized parameters, such as optimal treatment parameters and intervention period, is needed in the future.

## Systematic review registration

[https://www.crd.york.ac.uk/prospero/display_record.php?recordID=267596], identifier [CRD42022354363].

## Introduction

Stroke is a major cause of death and disability across the world. According to the Global Burden of Disease (GBD) report, stroke remained the second-leading cause of death of total deaths and the third-leading cause of death and disability worldwide [1]. Moreover, studies have shown that stroke is the leading cause of long-term disability in stroke patients from the United States, Japan, and France [2]. The fatality rate is significantly lower than before with the progress and development of stroke treatment. However, 80% of the survivors have severe sequelae, and the disability rate is about 75% [3]. There is a decline in the quality-of-life of patients and aggravation of disease conditions. Therefore, improving upper and lower limb motor function of stroke patients through active and effective rehabilitation treatment is essential.

Several current interventions are used to improve upper and lower limb function, including mirror therapy (MT) [4], constraint-induced movement therapy [5], acupuncture [6], afferent stimulation [7] and robot-assisted therapy [8]. However, these interventions have some shortcomings such as poor repeatability, boring, high cost and difficult evaluation [6, 9]. Electromyographic biofeedback (EMG-BFB) therapy has the advantages of high safety, timely intuitive feedback, and objective evaluation [10].

Biofeedback (BFB) has been used in rehabilitation for more than 40 years [11]. The technique tries to transduce a patient's physiological electrical response-which would typically be subliminal-into an aural or visual stimulus, in terms of limb function recovery after stroke this is normally in the form of an electromyogram. This records a difference in potential along the length of a muscle using electrodes over the skin surface. After amplification, the signal is displayed in a simplified format to the patient, for example, by hearing a change in auditory tone or by seeing a response on an oscilloscope display [12]. BFB can be used to examine muscle activation patterns or as a tool to support and improve various neuromuscular rehabilitation methods [13]. The EMG-BFB ability to record muscle activity even when there is no perceptible or discernible movement broadens the EMG-BFB potential application areas [14].

EMG-BFB therapy is currently regularly used in clinical settings, but the effectiveness of this intervention is still under debate. A meta-analysis including eight randomized controlled

trials(RCTs) showed that EMG-BFB therapy was superior to the conventional therapies in strengthening the ankle dorsiflexion in stroke patients [15]. However, a systematic review by Woodford H et al. showed there were no statistically significant improvements of EMG-BFB therapy in the range of motion in the ankle, knee and wrist joints in stroke patients. However, this study showed that EMG-BFB was able to significantly improve shoulder range of motion [9]. And EMG-BFB therapy was cited in the Royal College of Physicians stroke guidelines [16], which concluded that there is currently no evidence of its superiority over conventional therapy.

Clinical studies in this technique have continued to be updated in recent years. Consequently, this meta-analysis and systematic review incorporates the latest clinical studies in this area, further focusing on the advantages and limitations of EMG-BFB in limb function after stroke.

## Methods

The protocol for this systematic review and meta-analysis was registered on PROSPERO (CRD42022354363). This study was reported according to the Preferred Reporting Items for Systematic Reviews and Meta-Analyses (PRISMA) guidelines [17].

### Search strategy

We carried out a systematic search in electronic databases, including PubMed, Embase, the Cochrane Library, and PEDro from inception to 1st May 2022. The search strategy included synonyms of major keywords and medical subject headings (MeSH) terms. The AND operator was used to combine the search results with regard to the target patients (stroke), intervention (electromyographic biofeedback therapy), and study design (randomized controlled trial) (The search detail was presented in S1 Table). And the reference lists of all relevant studies in previous reviews were checked to avoid potential omissions.

### Inclusion criteria

Studies matching the following criteria were entered in the meta-analysis:

(1) Participants: Patients with hemiplegia after stroke (cerebral infarction or cerebral hemorrhage) met the diagnostic criteria of stroke revised by the fourth National Cerebrovascular Academic Conference in 1996 by clinical diagnosis and CT or MRI examination; (2) Interventions: EMG-BFB therapy; Techniques were anticipated to involve placing electrodes on the skin surface above the muscle region being evaluated, and then showing the patient either a visual or audio representation of the electrical activity. We kept track of every change in the muscle areas employed, the types of biofeedback used, and the length of the recipient's therapy. (3) Comparison: the interventions administered to the control groups could be placebos, sham stimulation, or conventional rehabilitation therapy;(4) Outcomes: The primary outcome was limb function after stroke, assessed by scales including Fugl-Meyer assessment (FMA), the Modifed Motor Assessment Scale(MMAS),etc. The secondary outcomes included a range of motion (ROM), surface electromyography values, and activities of daily living (ADL); (5) Studies: we limited the eligible study type to randomized control trials for reliable results.

### Data extraction

Two reviewers (R.W. and S.Z.) independently screened the titles and selected relevant studies according to the abstracts. The eligibility of the screened articles was checked by a full-text

review, A third reviewer (J.L.) was consulted if any discrepancies arose. Sociodemographic features, clinical information, study characteristics, and intervention parameters were collected using a pre-specified form. The outcome data of limb function were also extracted from the main text, tables, and supplementary materials. And the two reviewers were cross-checked after the extraction was completed.

### Risk of bias

The risk of bias for all the included studies was assessed by two independent reviewers (R.W. and S.Z.) using the Cochrane Collaboration's tool (Risk of Bias version 2 [RoB2], London, United Kingdom) [18]. Discrepancies between the two reviewers were mediated by a third reviewer (J.L.). Judgments for the overall risk of bias according to the algorithms provided by the RoB2 tool were calculated automatically based on five bias domains.

### Statistical analysis

Meta-analyses were conducted using Review Manager (RevMan) software version 5.4 (Cochrane, Copenhagen, Denmark). All data evaluated for quantitative merging were continuous outcomes from mean change values and their standard deviations. Standard deviation values were estimated for trials that reported pre- and post-therapy scores without change values, and the data were imputed using the Cochrane Handbook for Systematic Reviews of Interventions [19]. Considering various limb function scales, we chose standardized mean differences (SMDs) and 95% confidence intervals (CIs) to evaluate effect sizes. Estimated SMDs were then pooled together using a random-effects model via the inverse variance method. The level of statistical heterogeneity was measured using the $I^2$ statistic, and substantial heterogeneity was defined if $I^2$ values > 50%. We performed sensitivity analyses in two ways, by excluding studies with a high risk of bias and by leave-one-out exclusion. The threshold for statistically significant effect size was set at $P < 0.05$.

Subgroup analyses were conducted to identify the sources of heterogeneity and inter-subgroup differences in effect sizes. Multiple subgroup categories were pre-specified as follows: (1) Analysis between scales of the same kind; (2) upper or lower limb; (3) and time of assessment.

Funnel plots and Egger's tests were used to assess the potential publication bias for outcomes evaluated in more than 10 studies. Publication bias was considered significant if the funnel plot was asymmetric or if the P-value of Egger's test was < 0.1.

## Results

We collected 6091 records from four databases and identified five records from other sources, including reference lists from previous relevant reviews. After removing duplicates, 4051 items remained. We identified 4051 records via title screening, and the abstracts of these records were screened more comprehensively. A total of 74 articles were selected for a full-text review, and ten articles were considered eligible for quantitative meta-analysis. The details of screen process are shown in Fig 1.

### Study characteristics

S2 Table shows a summary of the features of the included studies. This meta-analysis including ten studies. Among these, two trials took place in the United States [20, 21], one in Asia [22], three in Turkey [23–25], one in the United Kingdom [26], one in Italy [27], one in Spain [28], and one in Iran [29].

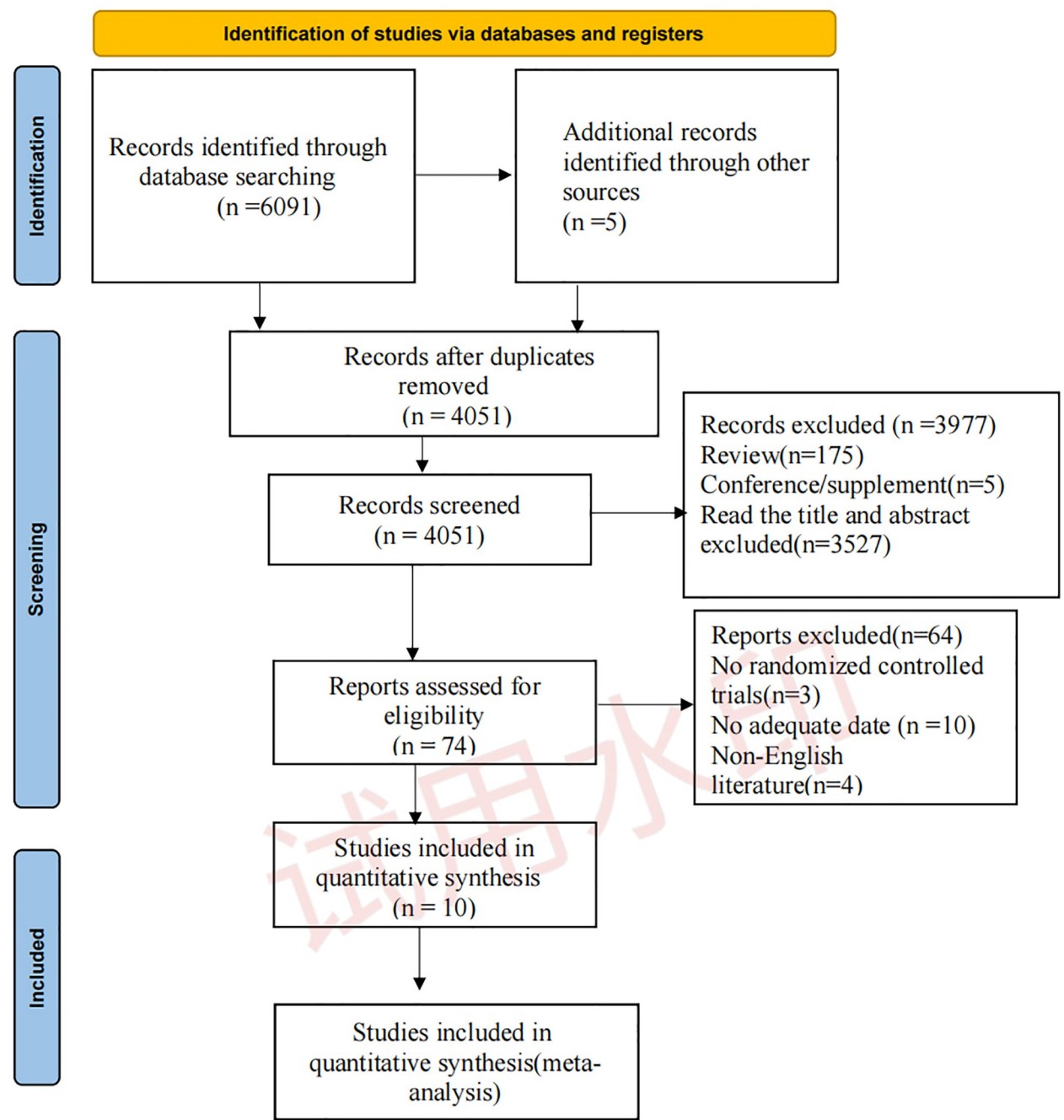

**Fig 1. PRISMA flow diagram depicting the study process.** N, the number of records; PRISMA, Preferred Reporting Items for Systematic Reviews and Meta-Analyses.

A total of 303 participants were enrolled; the number of participants in each study varied from n = 5 to n = 22. The mean age of participants in each group ranged between 45 and 67.4 years, and the proportion of women ranged from 22.2 percent to 77.8 percent, the proportion of right-sided hemiplegia in stroke patients fluctuates between 40 and 100 percent, and two studies have not reported on this aspect. It is worth noting that there was an extensive variation between studies for the time of mean stroke duration of the participants (ranging from 6 to

199.3 days). Of the 10 included studies, five studies included patients with lower limbs, and the other five studies included patients with upper limbs. The duration of EMG biofeedback treatment for patients included in the study fluctuated from two weeks to three months. Interventions were given for approximately 10–30 days, both five times a week for 20 to 60 minutes, according to the included trials. Most of the articles did not emphasize the time of a stroke of the included patients; only one article emphasized that the time of the included patients was more significant than or equal to 1 year for patients with chronic stroke.

## Risk of bias

The risk of bias for individual studies and the percentage graph depicting the risk of bias summary are shown in S3 Fig. Among the 10 included studies, a high risk of overall bias was detected in three investigations, and two studies were rated as low risk. The remaining five studies (50%) exhibited some concerns regarding overall bias. With respect to risk of bias domains, three studies reported a high risk of bias in the randomization process [24, 26, 29]. The majority of the studies exhibited a low risk of bias in the three evaluated domains, namely, missing outcome data, outcome measurement, and selective reporting.

## Effects of EMG-BFB in post-stroke limb dysfunction

Most studies used several functional scale tests related to FMA, hand function, etc. to measure limb impairment following stroke. The meta-analysis of this primary outcome showed that EMG-BFB therapy was associated with significantly improved post-stroke upper and lower limb function (SMD, 0.44; 95% CI, 0.12–0.77; P = 0.008) as compared with the control group (Fig 2). The heterogeneity of the pooled results was acceptable ($I^2$ = 47%, P = 0.05). Leave-one-out sensitivity analyses were performed, and found that heterogeneity was significantly

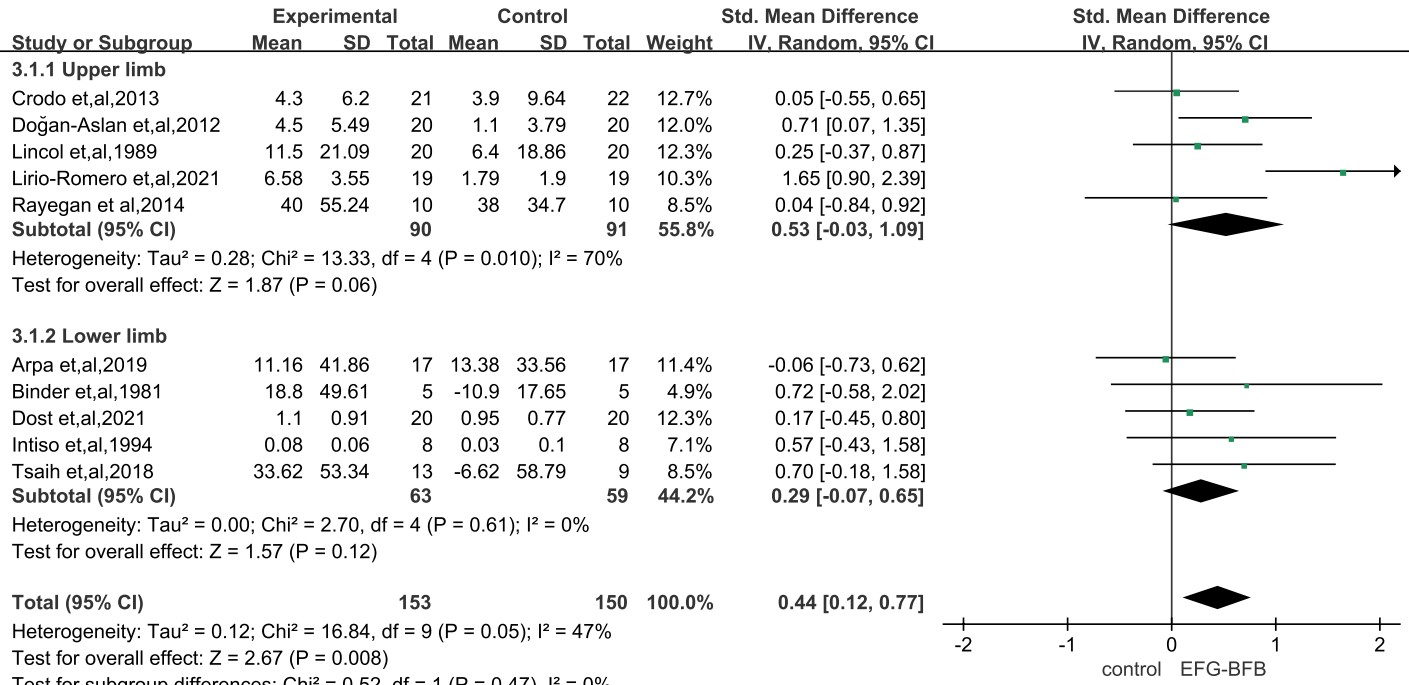

**Fig 2. Pooled effects of changes in upper or lower limb function after electromyographic biofeedback therapy as compared to control interventions.** SMD, standardized mean difference. Weights are from random-effects model.

reduced by excluding studies [28] ($I^2$ = 0%; P = 0.73) (Table 1). Sensitivity analyses that excluded high-risk studies were also performed ($I^2$ = 53%; P = 0.005) (S1 Fig); No statistically significant differences were found between the results of the two sensitivity analyses, indicating stable results. For detailed results are shown in S1 and S2 Figs.

## Subgroup analyses

S4 Fig shows the effect of different scales to evaluate the effect of EMG-BFB on post-stroke patients and the existing data do not prove the role of EMG-BFB in upper and lower limb motor function scale (FMA) (SMD, 0.78; 95% CI, -0.10–1.66; P = 0.08), hand function (SMD, 0.18; 95% CI, -0.03–0.69; P = 0.49), etc., nor does it prove the role in improving muscle strength changes(SMD, 0.26; 95% CI, -0.47–0.99; P = 0.48) and velocity(SMD, 0.63; 95% CI, -0.17–1.42; P = 0.12) after disabling stroke. Fig 3 reports on the effects of different function scales on the improvement of upper or lower limb function after stroke, and the results are not statistically significant (Upper limbs: SMD, 0.53; 95% CI, -0.03–1.09; P = 0.06; Lower limbs: SMD, 0.29; 95% CI, -0.07–0.65; P = 0.12). However, a subgroup analysis according to time of assessment showed the effect sizes of short-term (less than one month) effect (SMD, 0.33; 95% CI, 0.02–0.64; P = 0.04) was significant, but the long-term (more than one month) was not (SMD, 0.61; 95% CI, -0.11–1.33; P = 0.10) (Fig 4).

## Secondary outcome measures

**Range of motion.** Five trials [21, 23–25, 28] used the range of motion through a joint as an outcome measure: three at the ankle, and one each for the glenohumeral, and wrist. Three trials [21, 24, 25] reported the treatment ≤ 4-week ankle active range, including a total of 84 patients, with no statistically significant difference in improvement in ankle motion between the trial and control groups (SMD 1.11, 95% CI -1.27 to 3.48, P = 0.36).However, the only trial assessing the glenohumeral ROM [28] after treatment for more than four weeks (SMD 1.49, 95% CI 0.77 to 2.22,P<0.0001)and wrist ROM [23] (SMD 0.77, 95% CI 0.13 to 1.42,P = 0.02) did suggest a beneficial effect (Fig 5).

**EMG values.** Three studies [23, 25, 27] measured the values of electromyogram (EMG). Two studies [25, 27] measured the EMG values of the anterior tibialis muscle, and the other [23] measured the EMG values of the upper limb. None of the three studies showed a significant effect on the combined results of EMG values (SMD, 0.89; 95% CI, -1.81 to 3.59;) (S5 Fig).

**Table 1. Summary of leave-one-out sensitivity analysis for the effects on limb function after stroke.**

| Study | SMD (95%CI) | Heterogeneity ($I^2$) | P-values |
|---|---|---|---|
| Binder et al, 1981 [21] | 0.43(0.09,0.78) | 52% | 0.01 |
| Cristina et al, 2021 [28] | 0.27(0.02,0.52) | 0% | 0.03 |
| Domenico et al, 1994 [27] | 0.44(0.08,0.79) | 52% | 0.02 |
| Gülseren et al, 2021 [25] | 0.26(-0.02,0.55) | 51% | 0.009 |
| Lincol et al, 1989 [26] | 0.47(0.10–0.85) | 52% | 0.01 |
| Meryem et al, 2012 [23] | 0.41(0.05,0.77) | 50% | 0.03 |
| Paul et al, 2013 [20] | 0.50(0.15,0.86) | 47% | 0.006 |
| Peih et al, 2018 [22] | 0.42(0.07,0.78) | 51% | 0.02 |
| Rayegan et al, 2014 [29] | 0.48(0.13,0.83) | 50% | 0.007 |
| Selcan et al, 2019 [24] | 0.51(0.16,0.85) | 46% | 0.004 |

SMD, standardized mean difference; CI, confidence interval.

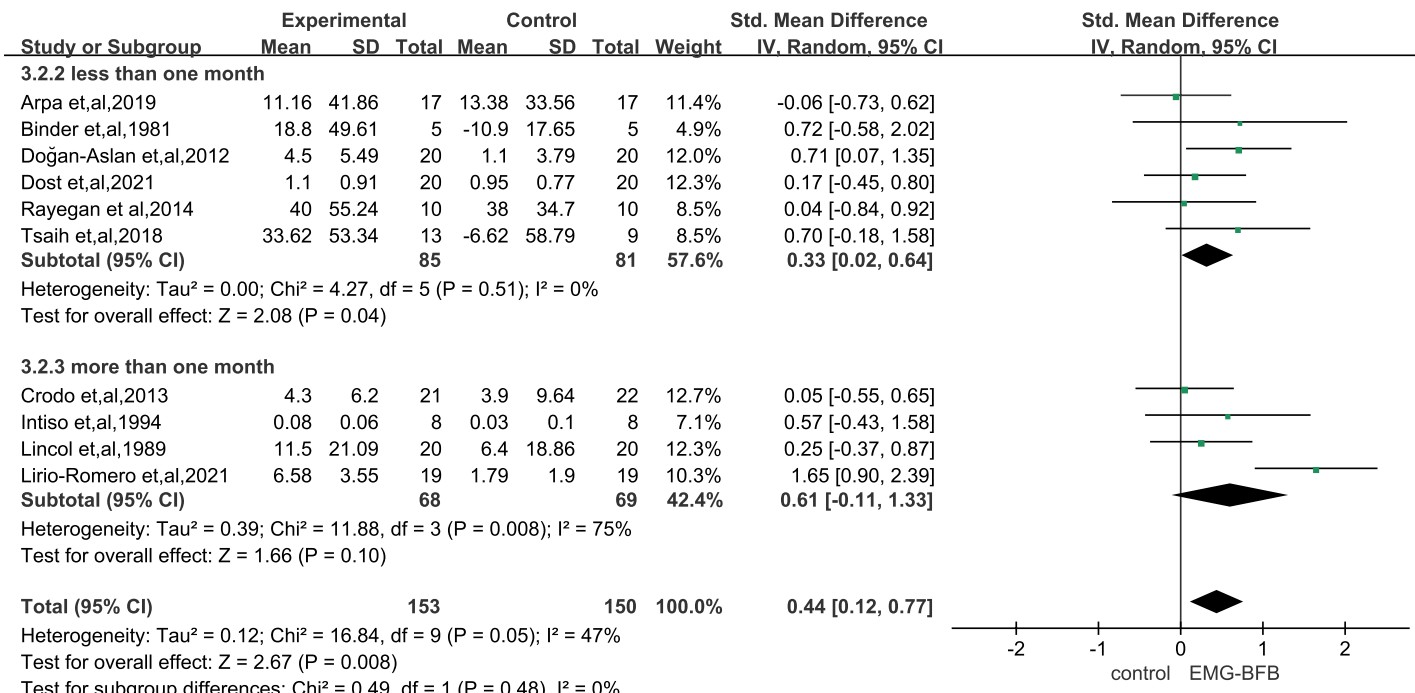

**Fig 3. Subgroup analysis assessed limb function improvement, stratified by assessment time point.** SMD, standardized mean difference. Tests for weight and heterogeneity between subgroups were derived from random effects models.

**Barthel index.** Four studies [20, 23, 24, 27] reported the post-stroke living index. There was no statistically significant improvement in the living index between the EMG-BFB and control groups (SMD, 8.13;95%CI, -4.35 to 20.62; $I^2$ = 66%) (S6 Fig). However, a sensitivity analysis excluding studies [20] with a high risk of bias demonstrated a beneficial effect (SMD, 13.39; 95% CI, 2.34–25.52; P = 0.02) that reduced the degree of heterogeneity ($I^2$ = 25%; P = 0.26) (S7 Fig).

**Publication bias.** Neither evaluations of the symmetry of the funnel plot (S8 Fig) nor the results of Egger's test (P = 0.4038) detected any significant publication bias across the 10 included studies. Therefore, we concluded that the probability that the effect size of EMG-BFB interventions was influenced by publication bias was low.

## Discussion

The results of this meta-analysis showed that the EMG-BFB therapy had a statistically significant improved on the upper and lower limb function scale, suggesting that EMG-BFB therapy can improve limb function after stroke. The mean age of the patients in each group ranged from 45 to 67.4 years. The relative young age may be one of the reasons for this conclusion. But the effect of different scales to evaluate the effect of EMG-BFB on post-stroke patients and the existing data do not prove the role of EMG-BFB in upper and lower limb motor function scale. In addition, the subgroup analysis by time of assessment supported the immediate effects of EMG-BFB therapy on limb function. However, the lasting efficacy of this modality might be limited. Moreover, according to the secondary outcome, we found among the five included studies, EMG-BFB therapy showed a statistically significant beneficial effect on wrist and shoulder range of motion [23, 28], but only one study supported the evaluation of wrist and

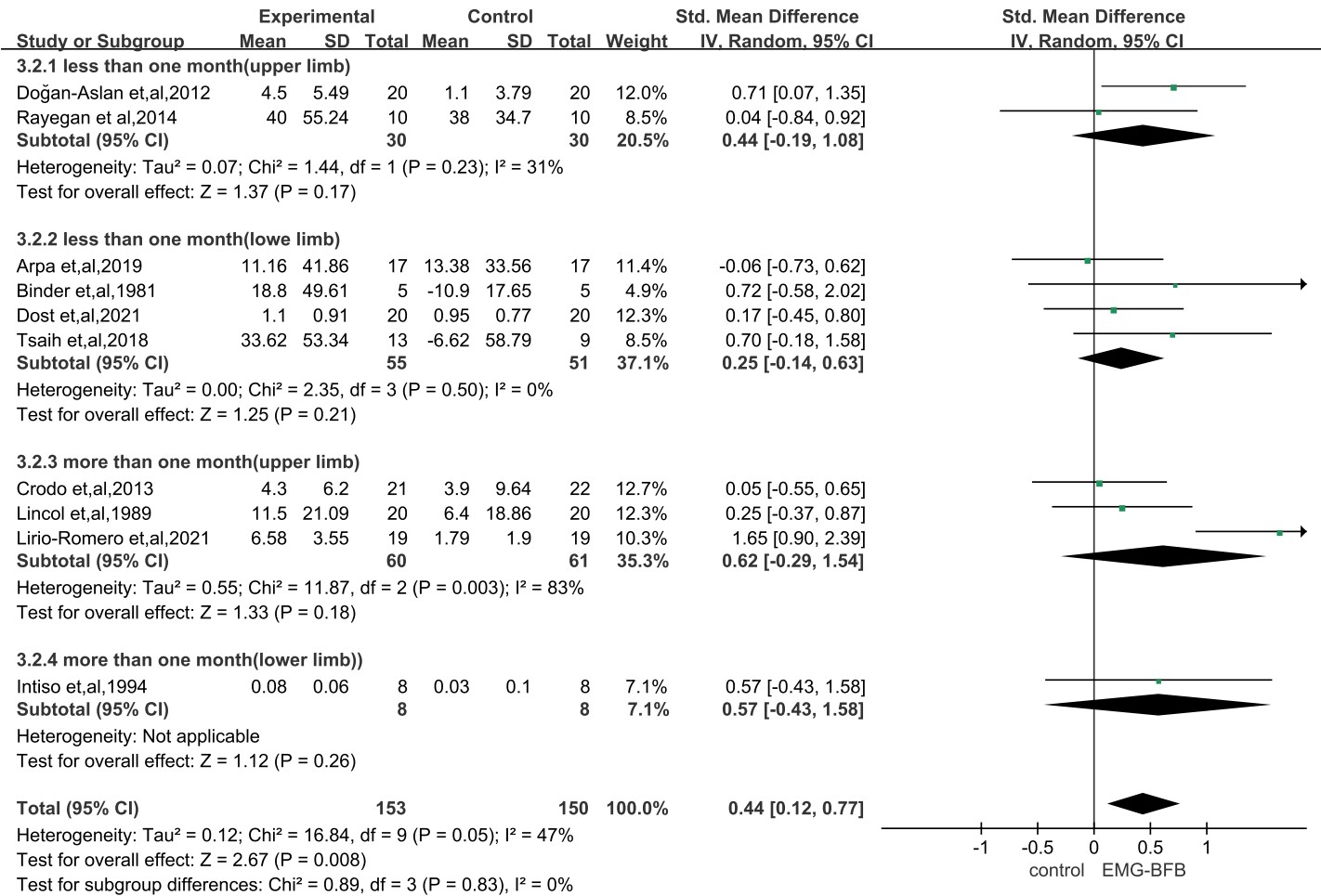

**Fig 4. Subgroup analysis assessed improvement in limb function in the upper or lower limb, stratified by assessment time point.** SMD, standardized mean difference. Tests for weight and heterogeneity between subgroups were derived from random effects models.

shoulder range of motion, and there is no evidence that EMG-BFB therapy can improve the effective changes in ankle range of motion, life index and EMG values after stroke.

Early review evaluated the impact of EMG-BFB on motor strength after stroke, but only one study included an assessment of motor strength, making the final meta-analysis impossible. However, the improvement in shoulder range of motion, a secondary outcome, was consistent with the findings of this study [9]. In addition, a previous meta-analysis evaluating EMG-BFB therapy to improve limb function after stroke only reported on the effect on the lower extremity [9]. In this study, the functions of the upper and lower limbs were reported separately in subgroup analysis. Although the effect on lower extremity function was not statistically significant, given the relatively small sample sizes of all included studies and the limited number of comparisons available for subgroup analysis, these variables need to be taken into account when interpreting the results.

An interesting phenomenon is that while the overall pooled results in our study are meaningful, the subscale results are not. This may be due to the different measurement focus of different scales and the small number of studies included in each subscale, which may lead to the bias of research results. More studies are needed to prove this uncertainty in the future.

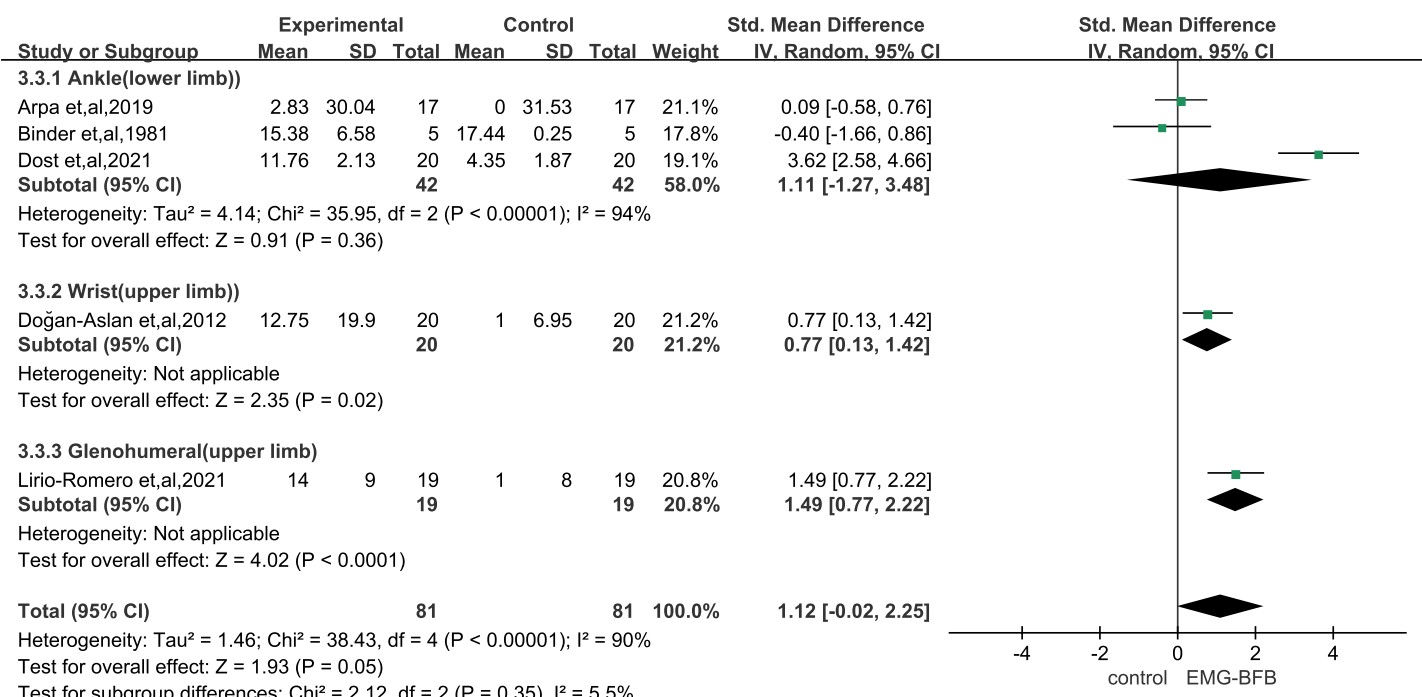

**Fig 5. Secondary outcome: Forest plot of range of motion.** SMD, standardized mean difference. Tests for weight and heterogeneity between subgroups were derived from random effects models.

Although heterogeneity between studies in this meta-analysis was considered, we did perform sensitivity analyses and we found heterogeneity was significantly decreased by excluding the [28], indicating that this study may be the source of its heterogeneity. This is likely due to the inclusion criteria for this study, which required all stroke patients with upper limb paralysis and a modified Ashworth scale spasticity score of 3 points or less. The intervention in this study is individualized, and the study developed individualized physiotherapy combined with EMG-BFB regimen for each participant. In addition, the age of the participants may also be an influencing factor, the mean age of the patients included in this study was 45 years. And it has been shown that brain plasticity decreases with ageing because of the gradual loss of internodes and myelin degeneration, which could make motor learning more difficult [30]. These may account for the source of their heterogeneity. After excluding the study, the effect of EMG-BFB therapy on the post-stroke upper and lower limb function improvement was still statistically significant, indicating that the meta-analysis results were robust.

The patient may not feel the "presence" of their limbs after a stroke since their mobility is limited, especially in the early post-stroke period [31]. However, EMG-BFB therapy can give patients visual cues to understand where limb muscle movement activation is occurring, assisting patients to begin training early, and providing helpful feedback for patients at the stage when the motor activation of limb muscles cannot be detected by other means. EMG-BFB therapy can also raise awareness of the new target muscle, then facilitate the separation of muscle activity, help patients establish a new movement pattern of cognitive requirements process, and guide patients in using a new movement pattern [31, 32]. In addition, with the establishment of new movement patterns and cortical reorganization after a successfully activating dissociation of muscle activity, valuable feedback can be provided to patients without activating their muscles, allowing further functional recovery of innervated muscles. And establish the

right movement patterns, exercises, repetitions, and a structured training program with appropriate biofeedback is necessary [33].

## Limitations

There remain several limitations to this study. The number of EMG-BFB therapy trials conducted in stroke patients are small, most studies included between 5 and 21 patients. The number of studies included in the subgroup analysis was also small. The overall effects and conclusions derived from subgroup analyses would be more statistically convincing with more EMG-BFB therapy trials, and we recommend that trials overcome this issue in the future. Second, The language of the included study was only English, the sample size was not large, and the inclusion criteria, intervention time, intensity, frequency and instruments used were different, which would affect the extrapolation of the meta-analysis results; We also note that the included studies were all efficacy observations before and after intervention, and there was a lack of follow-up and safety evaluation of the long-term efficacy of the study outcomes; In addition, We also note that the variety of limb function scales evaluated in this meta-analysis might introduce measurement heterogeneity, although we calculated SMD values to reduce the potential influence of this concern. However, due to the differences in data indicators included in our study, it cannot be further verified by methods such as GRADE. Although some of the included studies also assessed balance and gait disorders, their influence was not considered in the moderator analyses because of incomplete data and should be further analyzed in the future. Last, the recovery and rehabilitation pathway in stroke depends on the stage of the disease. Overwhelming evidence suggests that the neuroplastic changes driven during the therapy depend on the stroke's chronic and the initial impairment level, however, we were unable to analyze this data because most of our included literature does not mention the stage and severity of stroke. Future studies could further investigate this issue.

## Conclusions

EMG-BFB therapy is an effective, noninvasive treatment that can improve upper and lower limb function in patients with hemiplegia after stroke. This meta-analysis suggests that short-term EMG-BFB therapy intervention is supported in improving upper and lower limb function in people with stroke hemiplegia, while the benefit of long-term follow-up appears to be limited. Moreover, the available data did not demonstrate the role of EMG-BFB therapy in restoring upper and lower limb function after stroke through specific upper and lower limb function capacity scales, nor in improving quality of life and related muscle electromyography values after stroke; The optimal parameters and population-specific effects of EMG-BFB on limb function severity and other modifiers after stroke require a rigorous approach and adequate sample size to inform the effective management of precise individualized EMG-BFB therapy in the future.

## Supporting information

**S1 Checklist. The PRISMA-checklist.**
(PDF)

**S1 Fig. Forest plot of sensitivity analysis by excluding studies with a high risk of bias.**
(DOC)

**S2 Fig. Risk of bias assessment for studies using the RoB2 tool risk of bias summary for individual studies.**
(DOC)

**S3 Fig. Percentage graph of risk of bias across domains.** Abbreviation: RoB2, Risk of Bias version 2.
(DOC)

**S4 Fig. Pooled effects of different limb function scales after electromyographic biofeedback therapy as compared with control interventions.**
(DOC)

**S5 Fig. Secondary outcomes: Forest plot of relevant muscle EMG values.**
(DOC)

**S6 Fig. Secondary outcomes: Forest plot of barthel index.**
(DOC)

**S7 Fig. Forest plot of subgroup sensitivity analysis by excluding studies with a high risk of bias.**
(DOC)

**S8 Fig. Funnel plot of scale scores based on the limb function evaluation index.**
(DOC)

**S1 Table. Literature search strategy in different electronic databases.**
(DOC)

**S2 Table. Demographic and social characteristics of the included studies.**
(DOC)

## Acknowledgments

We thank Dr. Kun Zhao for his advice and language correction in this paper.

## Author Contributions

**Conceptualization:** Shuangshuang Zhang, Juebao Li.

**Methodology:** Qifeng Tong.

**Software:** Jie Zhang.

**Visualization:** Xiangming Ye, Kai Wang.

**Writing – original draft:** Rui Wang.

**Writing – review & editing:** Kai Wang, Juebao Li.

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
