## [Decision Letter · Decision Letter 0]

10 Apr 2023

PONE-D-23-04219Electromyographic biofeedback therapy for improving limb function after stroke: A systematic review and meta-analysis.PLOS ONE

Dear Dr. li,

Thank you for submitting your manuscript to PLOS ONE. After careful consideration, we feel that it has merit but does not fully meet PLOS ONE’s publication criteria as it currently stands. Therefore, we invite you to submit a revised version of the manuscript that addresses the points raised during the review process.

We look forward to receiving your revised manuscript.

Kind regards,

Paraskevopoulos Eleftherios

Academic Editor

PLOS ONE

Additional Editor Comments:

Dear authors

The reviewer completed the evaluation of your manuscript. Reviewer recommends major revisions prior to resubmission. Please review the comments and resubmit your revised manuscript

Reviewers' comments:

Reviewer's Responses to Questions

**Comments to the Author**

1. Is the manuscript technically sound, and do the data support the conclusions?

Reviewer #1: Yes

2. Has the statistical analysis been performed appropriately and rigorously? 

Reviewer #1: Yes

3. Have the authors made all data underlying the findings in their manuscript fully available?

Reviewer #1: Yes

4. Is the manuscript presented in an intelligible fashion and written in standard English?

Reviewer #1: Yes

5. Review Comments to the Author

Reviewer #1: Please see attachment.

Thank you for your work on preparing this manuscript and for providing me with the chance to review it.

The current systematic review attempts to summarize the literature on the use of electromyographic biofeedback (EMG-BFB) therapy in the rehabilitation of upper and lower limb function following stroke. A total of 10 randomized controlled trials were included in the review and meta-analysis. Reported results show a significant improvement in “limb” function following the use of EMG-BFB therapy. There were no significant differences in improvements in upper or lower limb functions separately.

While the effort undertaken by the authors to conduct the literature search and provide such comprehensive qualitative and quantitative summary of evidence is very obvious, several issues limit the usability and comprehension of the results of this work as outlined below.

1.

The primary result of the analysis in the current study shows a significant improvement in limb function following the use of EMG-BFB therapy (line 210 and figure 2). However, analysis of improvements in upper and lower limbs separately show non-significant findings (line 229 and figure 3).

It remains to be explained why did the authors see the need to combine the analysis for upper and lower limbs, which could have shown the significant improvement by a mere increase in sample size. In clinical practice, therapists aim to see improvements in either upper limb function or lower limb function. Combining both limbs in the analysis is not justified.

2.

The authors reported that the final synthesis of the systematic review and meta-analysis included 10 studies, the details of which are provided in supplementary table 2. However, several statements throughout the Results and Discussion sections seem to refer to studies not listed among those 10, for example, the majority of studies mentioned in the paragraph starting at line 175 (e.g., Crow et al, Cordo et al, Tsaih et al…), also, Lirio-Romero et al (line 214), Dogan-Aslan et al (line 237), Arpa and Ozcakir (line 237), just to name few.

This leaves the reader wondering as to the actual number of studies included in the review and which ones they are.

3.

The search strategy included reference lists of all relevant studies in previous reviews (line 112). In the Introduction, two similar meta-analyses of randomized controlled trials on EMG-BFB therapy in patients with stroke are cited; one by Moreland et al, and the other by Woodford et al (line 89). It appears that not all studies included in those two meta-analyses are included in the current manuscript.

For example, the paper by Moreland included 8 randomized controlled trials on EMG-BFB therapy and lower limb function.

Also, the paper by Woodford included studies on range of motion in the ankle, knee, shoulder, and wrist joints.

4.

This systematic review was limited to randomized controlled trials of EMG-BFB therapy reporting changes in upper and lower limb function in post-stroke patients. While randomized trials are known to be the gold-standard for establishing cause-and-effect relationships between interventions and outcomes, the authors repeatedly stated that the objective and findings of this study showed the association between EMG-BFB therapy and upper and lower limb dysfunction (example, lines 29, 39, 274…).

The wording should be changed throughout the manuscript to reflect cause-and-effect not association.

5.

Among the results highlighted in the abstract (lines 44 – 46), the authors mention that EMG-BFB therapy caused significant improvement in the range of motion of glenohumeral and wrist joints. Further reading in the Results section (line 245) reveals that this statement is actually based on one study, meaning that, in accurate terms, this is not based on a meta-analysis.

The authors should either clarify this in the abstract or remove it from the abstract and keep it in the Results and Discussion sections where the reader can have a clear and accurate understanding of how this finding was reached.

Actually, the meta-analysis by Woodford et al (see comment #3 above) included randomized controlled trials that examined the range of motion at the shoulder and wrist joints which are not included in the current analysis. Including these studies, if applicable, will allow running a meta-analysis on those results.

In addition, this discussion of improvement in the range of motion does not mention whether it is active or passive range that showed the improvement.

6. PLOS authors have the option to publish the peer review history of their article (what does this mean?). If published, this will include your full peer review and any attached files.

Reviewer #1: No

---

## [Author Response · Author response to Decision Letter 0]

13 Jul 2023

Reply to Reviewer 

Major comments:

Comments 1：The primary result of the analysis in the current study shows a significant improvement in limb function following the use of EMG-BFB therapy (line 210 and figure 2). However, analysis of improvements in upper and lower limbs separately show non-significant findings (line 229 and figure 3). It remains to be explained why did the authors see the need to combine the analysis for upper and lower limbs, which could have shown the significant improvement by a mere increase in sample size. In clinical practice, therapists aim to see improvements in either upper limb function or lower limb function. Combining both limbs in the analysis is not justified.

Response 1: We appreciate the reviewer insightful suggestion and agreed that combining both limbs in the analysis is not justified. Therefore, we delete the figure 2, which represents changes in overall limb function and we changed the figures combining the upper and lower limbs to discuss the changes in upper and lower limb function due to electromyographic biofeedback separately and to distinguish between upper and lower limbs for both subgroups and secondary outcome measures. For details, see Figs. 2, 3,and 4. And it has been changed again in the abstract and discussion section.

Comments 2：The authors reported that the final synthesis of the systematic review and meta-analysis included 10 studies, the details of which are provided in supplementary table 2. However, several statements throughout the Results and Discussion sections seem to refer to studies not listed among those 10, for example, the majority of studies mentioned in the paragraph starting at line 175 (e.g., Crow et al, Cordo et al, Tsaih et al…), also, Lirio-Romero et al (line 214), Dogan-Aslan et al (line 237), Arpa and Ozcakir (line 237), just to name few. This leaves the reader wondering as to the actual number of studies included in the review and which ones they are. 

Response 2:We thank the reviewer for pointing this out. Ten studies were finally included in this meta-analysis. When the information was included, we used the surname of the research author as the index of the abstract, while the name of the research author was used as the index when the results and discussion were cited. In order to avoid confusion, we have unified the format and changed all the studies to the name of the research author, as can be found in Supplementary Table 2.

Comments 3：The search strategy included reference lists of all relevant studies in previous reviews (line 112). In the Introduction, two similar meta-analyses of randomized controlled trials on EMG-BFB therapy in patients with stroke are cited; one by Moreland et al, and the other by Woodford et al (line 89). It appears that not all studies included in those two meta analyses are included in the current manuscript. For example, the paper by Moreland included 8 randomized controlled trials on EMG-BFB therapy and lower limb function. Also, the paper by Woodford included studies on range of motion in the ankle, knee, shoulder, and wrist joints. 

Response 3:We thank the reviewer for pointing this out. Moreland et al. retrieved CINAHL database and finally included 8 studies. We carefully read the relevant studies, and finally did not include all the lstudies because some studies could not find complete articles or could not find complete data after finding them. And, the inclusion criteria of the study conducted by Woodford et al. are not only randomized controlled trials, but also quasi‐randomised studies. For this part of the studies, we did not fully include them. The specific inclusion and exclusion criteria are also explained in detail in the flow diagram.

Comments 4：This systematic review was limited to randomized controlled trials of EMG-BFB therapy reporting changes in upper and lower limb function in post-stroke patients. While randomized trials are known to be the gold-standard for establishing cause-and-effect relationships between interventions and outcomes, the authors repeatedly stated that the objective and findings of this study showed the association between EMG-BFB therapy and upper and lower limb dysfunction (example, lines 29, 39, 274…). The wording should be changed throughout the manuscript to reflect cause-and-effect not association. 

Response 4:We appreciate the reviewer insightful suggestion and agreed that the wording should be changed throughout the manuscript to reflect cause-and-effect not association. So we changed all the words about association in the text to cause-and-effect. See lines 29,38,43,283.

Comments 5：Among the results highlighted in the abstract (lines 44 – 46), the authors mention that EMG-BFB therapy caused significant improvement in the range of motion of glenohumeral and wrist joints. Further reading in the Results section (line 245) reveals that this statement is actually based on one study, meaning that, in accurate terms, this is not based on a meta-analysis. The authors should either clarify this in the abstract or remove it from the abstract and keep it in the Results and Discussion sections where the reader can have a clear and accurate understanding of how this finding was reached. Actually, the meta-analysis by Woodford et al (see comment #3 above) included randomized controlled trials that examined the range of motion at the shoulder and wrist joints which are not included in the current analysis. Including these studies, if applicable, will allow running a meta-analysis on those results. In addition, this discussion of improvement in the range of motion does not mention whether it is active or passive range that showed the improvement. 

Response 5: We appreciate the reviewer insightful suggestion and agreed that the significant improvement in glenohumeral and wrist range of motion with EMG-BFB treatment is based on one study. However, according to the results of our study in Figure 5, it can be found that the EMG biofeedback therapy has a significant improvement in the shoulder and wrist joints, so we have modified the results in the abstract. In addition, regarding the issue of improvement in the range of motion does not mention whether it is active or passive range that showed the improvement , we re-read the relevant studies and find that it was the change of active joint range of motion, so we also added relevant instructions in the article. See lines 42-45.

Minor comments: 

Comments 1:While the results reported show a significant effect size for short-term effects but not long-term ones, it is not mentioned anywhere in the text what is the timeframe meant by short- or long-term. Only in figure 4 it is mentioned that the demarcation is made by the 1-month mark. This should be clearly stated in the manuscript.

Response 1:We appreciate the reviewer insightful suggestion and agreed that the time frame meant by short- or long-term should be clearly stated in the manuscript. Therefore, we have added time frame in the abstract and results sections. See lines 48,242.

Comments 2:The Introduction can be further enriched with more information about the current knowledge on the effectiveness of EMG-BFB. Also, the Methods section (e.g., line 119) should include more details about the methodology of EMG-BFB. Alternatively, such details can be added about the methodology of EMG-BFB used in the included studies in Supplementary Table 2.

Response 2:We thank the reviewer for pointing this out. We further enriched our knowledge of the effectiveness of EMG-BFB in the introduction section as follows: BFB can be used to examine muscle activation patterns or as a tool to support and improve various neuromuscular rehabilitation methods.The EMG-BFB ability to record muscle activity even when there is no perceptible or discernible movement broadens the EMG-BFB potential application areas. See lines 83-88.

And We have added details about the EMG-BFB method in the Methods section as follows: Techniques were anticipated to involve placing electrodes on the skin surface above the muscle region being evaluated, and then showing the patient either a visual or audio representation of the electrical activity. We kept track of every change in the muscle areas employed, the types of biofeedback used, and the length of the recipient's therapy. See lines 123-127.

We also documented the timing and frequency of EMG biofeedback treatment in detail in Supplementary Table 2.

Comments 3:While this paragraph lists current interventions that are used and highly investigated to improve upper and lower limb function following stroke, all references cited are old. 

Response 3:We thank the reviewer for pointing this out. We acknowledge that all references cited are old, so we have corrected this error. Changed to relatively new literature. See lines 69-71.

Comments 4:Please provide a reference for this statement on interventions shortcomings；

Response 4: We have provide a reference for this statement on interventions shortcomings；See line 73.

Comments 5:The reference by the Royal College of Physicians (2004) is not listed in the References. Also, newer guidelines (2016) are available. 

Response 5: We thank the reviewer for pointing this out. We carefully read the newer guidelines and cited them in the references of our study. See line 98.

Comments 6:Why “more than 10 studies” when the current study included only 10? 

Response 6:This “more than 10 studies” refers to funnel plots and Egger's test generally used to assess the potential publication bias of results from more than 10 studies. It does not mean that more than 10 studies were included in this study.

Comments 7:The statement says “two trials took place in the US,” however references listed are three.

Response 7: We apologize for this error and thank the reviewer for pointing this out. We have revised the references listed to two trials. See line 184-185.

Comments 8:The mean age of participants in each group ranged between 45 and 67.4 years. This age represents relatively young patients in the included studies, a point which should be addressed in the Discussion section.

Response 8:We appreciate the reviewer insightful suggestion and agreed that relatively young patients should be addressed in the Discussion section. So we added a note in the discussion section. See lines 283-285.

Comments 9:Please double check the following sentence: “Regarding the etiology of dementia, five studies included patients with lower limbs, and the other five studies included patients with upper limbs.”

Response 9:We apologize for this error and thank the reviewer for pointing this out. We have made the following corrections :Of the 10 included studies, five studies included patients with lower limbs, and the other five studies included patients with upper limbs. See lines 196-198.

Comments 10:What is the unit for the doses of interventions? Re-write the sentence.

Response 10:The unit for the doses of interventions was day. And we have Re-write the sentence. See lines 198-200.

Comments 11:What about the risk of bias in deviations from intended interventions?

Response 11:The risk of bias from an intended intervention was divided into two scenarios according to the purpose of the study: one was to study the effect of intervention allocation, and the other was to study the effect of intervention adherence.

Comments 12:In the sentence “results are not clinically significant,” this should be changed to statistically significant.

Response 12: We apologize for this error and thank the reviewer for pointing this out. We've made a change. See line 238.

Comments 13:Please double check the wording in the sentence: “the probability that the effect size of phototherapy interventions was…” 

Response 13:We apologize for this error and thank the reviewer for pointing this out. We have made the following changes: Therefore, we concluded that the probability that the effect size of EMG-BFB interventions was influenced by publication bias was low. See lines 278-279.

Comments 14:Please double check that five studies showed a statistically significant beneficial effect on wrist and shoulder range of motion. 

Response 14:Range of motion was assessed in a total of five studies, three of which studied the ankle, one on the wrist and one on the shoulder, in which the change in ankle range of motion was not statistically significant and there was a significant beneficial effect on ankle and shoulder range of motion.

Comments 15:Please cite the sentence: “a previous meta-analysis evaluating EMGBFB…”

Response 15:We have added references to the study. See line 207.

Comments 16:Please provide a reference for this statement on impaired sensation early post stroke.

Response 16:We have provide a reference for this statement on impaired sensation early post stroke. See line 330.

Comments 17:Legends for figures 2 and 3 are switched.

Response 17:We apologize for this error. We have corrected this error. Because we have modified the original Figures 2-4, the relevant legend descriptions have also been changed.

Comments 18:Last box should say “studies included in quantitative synthesis (meta-analysis)” not qualitative

Response 18:We thank the reviewer for pointing this out. It has been modified to quantitative analysis. See figure 1.

Comments 19:Please double check the numbers under stroke side in the control group in the study by Cristina et. al. The control group has 19 patients but numbers for stroke side are mentioned for 8 patients only.

Response 19:We apologize for this error and we have corrected this error. The correct side of the stroke was 19 patients, 11 on the right side and 8 on the left side. See Supplementary Table 2.

Language comments:

Thank you very much for the reviewer's comments on the language modification of my manuscript. I have revised it one by one as required and marked it with red letters in the text.

Comments 1:Stroke is the leading cause of long-term disability in stroke patients… 

Review 1:Stroke is a major cause of death and disability across the world. See line 56.

Comments 2:EMG-BFB therapy now widely used in clinical practice… 

Review 2:EMG-BFB therapy is currently regularly used in clinical settings. See line 89.

Comments 3:In contrast, showed significant associations with improved… 

Review 3: However, this study showed that EMG-BFB was able to significantly improve shoulder range of motion. See line 96.

Comments 4:Regarding the etiology of dementia, five studies included patients with lower limbs, and the other five studies included patients with upper limbs.

Review 4:Of the 10 included studies, five studies included patients with lower limbs, and the other five studies included patients with upper limbs. See line 196-198.

Comments 5:The mean EMG-BFB frequencies and duration of the participants ranged from… 

Review 5:The duration of EMG biofeedback treatment for patients included in the study fluctuated from two weeks to three months. Interventions were given for approximately 10-30 days, both five times a week for 20 to 60 minutes, according to the included trials.See line 198-200.

Comments 6:We found by excluding studies (Lirio-Romero et al, 2021) significantly reduced… 

Review 6:and found that heterogeneity was significantly reduced by excluding studies See line 222.

Comments 7:This is a very long sentence. Please re-write. 

Review 7:However, EMG-BFB therapy can give patients visual cues to understand where limb muscle movement activation is occurring, assisting patients to begin training early, and providing helpful feedback for patients at the stage when the motor activation of limb muscles cannot be detected by other means. EMG-BFB therapy can also raise awareness of the new target muscle, then facilitate the separation of muscle activity, help patients establish a new movement pattern of cognitive requirements process, and guide patients in using a new movement pattern.See line 331-337.

Comments 8:....with most patients between 5 and 21 patients. 

Review 8:.....most studies included between 5 and 21 patients. See line 346.

Comments 9:And due to the various scores and scales analysed were diferent and 

expressed in means, there is no method used to qualify the evidence, like 

GRADE.

Review 9:However, due to the differences in data indicators included in our study, it cannot be further verified by methods such as GRADE. See line 359.

We would like to thank the editor and reviewers again for taking the time to review our manuscript. We hope this revised manuscript has addressed yours concerns，and look forward to hearing from you.

Yours sincerely.

Juebao Li

25 May, 2023

Research Center for Rehabilitation Medicine, Rehabilitation & Sports Medicine Research Institute of Zhejiang Province, Department of Rehabilitation Medicine, Zhejiang Provincial People's Hospital, Affiliated People's Hospital, Hangzhou Medical College, Hangzhou, Zhejiang, China

---

## [Editor Report · Decision Letter 1]

21 Jul 2023

Electromyographic biofeedback therapy for improving limb function after stroke: A systematic review and meta-analysis.

PONE-D-23-04219R1

Dear Dr. li,

We’re pleased to inform you that your manuscript has been judged scientifically suitable for publication and will be formally accepted for publication once it meets all outstanding technical requirements.

Kind regards,

Paraskevopoulos Eleftherios

Academic Editor

PLOS ONE
---

## [Editor Report · Acceptance letter]

7 Aug 2023

PONE-D-23-04219R1 

Electromyographic biofeedback therapy for improving limb function after stroke: A systematic review and meta-analysis. 

Dear Dr. Li:

I'm pleased to inform you that your manuscript has been deemed suitable for publication in PLOS ONE. Congratulations! Your manuscript is now with our production department. 

Kind regards, 

on behalf of

Dr. Paraskevopoulos Eleftherios 

Academic Editor

PLOS ONE